# Neuropsychological Outcome of Critically Ill Patients with Severe Infection

**DOI:** 10.3390/biomedicines10030526

**Published:** 2022-02-23

**Authors:** Maria Della Giovampaola, Irene Cavalli, Luciana Mascia

**Affiliations:** 1Department of Medical and Surgical Sciences, University of Bologna, 40126 Bologna, Italy; mar.dellagiovampaola@studio.unibo.it (M.D.G.); irene.cavalli@studio.unibo.it (I.C.); 2Department of Biomedical and Neuromotor Sciences, University of Bologna, 40126 Bologna, Italy

**Keywords:** sepsis, septic shock, sepsis-associated encephalopathy, sepsis-associated long-term cognitive impairment, COVID-19

## Abstract

Sepsis and septic shock represent important burdens of disease around the world. Sepsis-associated neurological consequences have a great impact on patients, both in the acute phase and in the long term. Sepsis-associated encephalopathy (SAE) is a severe brain dysfunction that may contribute to long-term cognitive impairment. Its pathophysiology recognizes the following two main mechanisms: neuroinflammation and hemodynamic impairment. Clinical manifestations include different forms of altered mental status, from agitation and restlessness to delirium and deep coma. A definite diagnosis is difficult because of the absence of specific radiological and biological criteria; clinical management is restricted to the treatment of sepsis, focusing on early detection of the infection source, maintenance of hemodynamic homeostasis, and avoidance of metabolic disturbances or neurotoxic drugs.

## 1. Introduction

Sepsis is one of the major causes of illness around the world [1,2]. Important efforts have been made in the past decades to improve its management, but the neurological and functional consequences of sepsis still remain poorly understood. Among the long-term organ dysfunctions, neurological impairment is one of the most debilitating, with great impact on the survivors’ quality of life. The aim of this paper is to review the neurological complications related to sepsis and septic shock, both in the acute phase and in the long term.

## 2. Epidemiology

Sepsis and septic shock represent global burdens of disease, affecting millions of people around the world each year, with a global incidence of sepsis-related deaths of 19.7% [1,2]. 

Sepsis-related complications represent a major healthcare problem. Sepsis-associated encephalopathy (SAE) is a relatively common cause of altered mental status in critically ill patients admitted to the ICU for severe infection, with an incidence between 8 and 70% according to diagnostic criteria [3]. Although mostly reversible, SAE is associated with higher mortality [4]. Among survivors, long-term cognitive impairment, affecting functional abilities, verbal learning and memory, quality of life, and the ability to return to work, have been reported [3,5].

## 3. Pathophysiology of Sepsis-Associated Encephalopathy

SAE is a severe brain dysfunction with a wide range of clinical presentations, from mildly altered mental status to deep coma [4,6,7].

The pathophysiology of SAE is multifactorial (Figure 1); in the absence of any evident direct infection of the brain, it is most likely related to the effect of systemic inflammation on the microglia and of hemodynamic impairment on cerebral perfusion [4,6,8,9,10,11].

### 3.1. Neuroinflammation

Microglia and endothelial activation, and dysfunction of the blood–brain barrier (BBB) and astrocytes, are the key components involved in neuroinflammation [12].

Afferent signals from the damaged tissue activate neuroinflammation, inducing microglial and endothelial activation [12,13]. Among the possible pathways, the nerve vagus detects the inflammatory intraperitoneal signal, carrying this information to the medullary autonomic nuclei. At this level, modulation of the response to sepsis begins [12]. Microglial cells have different roles in brain homeostasis, depending on which phenotype is expressed, anti-inflammatory (M2) or proinflammatory (M1) [10].

During sepsis, activation of the microglia is mostly proinflammatory, and it is implicated in neuronal apoptosis, and synaptic and neurochemical disturbances. Neuroinflammation predisposes to the alteration of the N-methyl-D-aspartate receptor, resulting in synaptic dysfunction, cell death, and functional consequences on cognition and behavior [12,14]. Aside from dopaminergic, β-adrenergic and GABAergic dysregulation during SAE, dysregulation of the cholinergic pathway plays a key role in the modulation of inflammation, delirium and long-term cognitive impairment [9,10,12,15,16]. The imbalance between the release and reuptake of neurotransmitters exposes neurons to excitotoxicity [10].

The altered integrity of the cerebral endothelial cells results in the breakdown of the BBB, with subsequent increased permeability to neurotoxic factors from the peripheral circulation [17]. The altered endothelium also produces microcirculatory impairment, characterized by alterations in vascular tone and activation in the coagulation cascade. This predisposes the brain to both ischemic and hemorrhagic lesions [12,13,18].

Microglial and endothelial activation, together with the disturbance of the BBB, enhance neuroinflammation; this final pathway has been proposed to cause the transition from SAE to cognitive impairment [3,9,13,17,18].

Finally, the pathogenesis of SAE has been linked to the induction of nitric oxide-producing enzymes, responsible for the pathological processes of neurons, and an increase in apoptosis.

### 3.2. Hemodynamic

Systemic hemodynamic instability during septic shock is a consequence of the systemic dysregulated host response to the infection, and it may be due to both macrocirculatory and microcirculatory dysfunction. Macrocirculation is affected by a reduction in the mean arterial pressure (MAP), consequent to a decrease in systemic vascular resistance (SVR). In addition, sepsis-induced myocardial dysfunction may occur, leading to a reduction in cardiac output [19]. On the other hand, microcirculation is impaired because of alterations in microcirculatory autoregulation, mitochondrial dysfunction and microvascular thrombosis; the latter may lead to disseminated intravascular coagulation [20,21,22].

Cerebral autoregulation is defined as a homeostatic process that maintains the cerebral blood flow (CBF) across a range of mean arterial pressures (MAP) between 60 and 160 mmHg [23,24]. Several mechanisms (myogenic, neurogenic, metabolic and endothelial) are involved in cerebral autoregulation [24], each of which may be impaired during sepsis, exposing the brain to global ischemia or brain edema, respectively [25,26,27].

In septic patients, the impairment of cerebral autoregulation has been investigated, demonstrating that its alteration is more common in early versus late sepsis, and in patients with SAE [26,27]. In an experimental model of sepsis [28], dynamic cerebral autoregulation and neurovascular coupling have been shown to be impaired, compared to non-septic animals. In a small cohort of 30 septic patients, impairment of cerebral autoregulation was reported in up to 60% of patients, mostly during day 1 [29]; this was associated with the development of sepsis-associated delirium. Recently, in a large observational prospective study [25], impairment in cerebral autoregulation, associated with the development of SAE, has been found in more than 50% of septic patients.

## 4. Clinical Manifestations and Diagnosis

A prevalence of 8–70% of SAE has been reported in critically ill patients [4]. This wide range may be explained by the non-specific clinical manifestations and the difficulties of providing a certain diagnosis. Several conditions, such as medications, electrolyte disturbances, and direct central nervous system infections (meningitis, encephalitis, cerebral abscess, and septic emboli), must be ruled out before a diagnosis of SAE can be made [7,12]. Therefore, a thorough investigation, including a complete physical and neurological examination, medication review, and appropriate instrumental diagnosis, in critically ill patients is the starting point for the detection of SAE.

Patients with SAE frequently present with an abnormal mental status, disturbances in the sleep–wake cycle, or evidence of hallucinations, restlessness, or agitation [4]. These symptoms resemble delirium; however, these two conditions are not synonymous, as SAE is one of the main causes of delirium, but delirium is not the only clinical presentation of SAE [7]. Other clinical features of SAE may include focal deficit, roving eye movements, asterixis, tremor, multifocal myoclonus, seizures, paratonic rigidity, and flexor or extensor posturing [30]. The routine use of CAM-ICU and ICDSC scales is mandatory to assess delirium in patients who present with alterations in behavior and mental status, although these evaluations may not be sensitive enough to detect the full spectrum of SAE [12,30]. Furthermore, since sedation is not considered in the scales used to assess delirium, it may represent a confounding factor in the evaluation of mental status. Therefore, daily assessment of the Richmond Agitation and Sedation Scale (RASS) score is strongly recommended [31].

Aside from a routine assessment of delirium, daily neurological examinations, including the level of arousal, brainstem function, and motor response, are fundamental in neuromonitoring. The Glasgow Coma Scale (GCS) or the Full Outline of Unresponsiveness (FOUR) scales are generally used for this purpose, especially in comatose patients [31].

### 4.1. Instrumental Diagnosis

EEG represents a sensitive tool for detecting brain electric abnormalities in several clinical scenarios, but it is not specific in the diagnosis of SAE [6,10]. When continuous EEG monitoring is used, it may detect electrical seizures and epileptiform pathways, especially if they are not associated with clinical manifestations [32].

During sepsis, a wide range of alterations in the EEG pattern have been described.

Progressive slowing of the normal alpha rhythm, appearance of theta to delta activity, triphasic waves, and malignant burst suppression in severe malignant encephalopathy [4,6,10] have been described in patients with SAE. These EEG patterns have been classified into four categories of increasing severity of SAE by Young et al. [32].

In a retrospective study [33], critically ill patients without a known acute neurologic injury underwent continuous EEG monitoring to identify any alterations in the EEG pattern. This study showed that patients with sepsis had higher rates of seizures and periodic epileptiform discharge than non-septic patients. Both types of EEG abnormalities were associated with death and poor outcomes at hospital discharge.

A prospective study [31] on a population of septic patients reported that the overall prevalence of non-convulsive seizures (NCS) and periodic discharge (PD) was around 25%. These findings confirmed previous studies, although the authors did not find any correlation between NCS and mortality. Conversely, they showed that a lack of EEG reactivity was associated with higher 1-year mortality.

Recently, a prospective study in patients with sepsis investigated the correlation between EEG abnormalities, mortality and the occurrence of delirium [34]. The presence of a delta-predominant background, absence of EEG reactivity, PD, Synek grade ≥ 3 and Young grade > 1 were independent predictors of ICU mortality, and were associated with the occurrence of delirium.

These abnormal findings provide objective evidence of altered cerebral electrical activity, and have been related to the severity of the underlying encephalopathy [10,35], although they are not specific to SAE [4,6,36,37].

Alteration in the latency or amplitude of the somatosensory-evoked potentials has been frequently reported in this patient population [30], and it is not influenced by sedative medications [10].

### 4.2. Biological Markers

Biomarkers of astrocytes and neuronal injury, such as neuron-specific enolase (NSE) and S-100 β-protein, have been reported to be elevated in patients with SAE [38]. Their usefulness in determining brain injury severity and in aiding prognosis is still contradictory [39]. A retrospective analysis of a sepsis cohort reported that an NSE concentration > 12.5 μg/L was independently associated with a 23% and 29% increased risk of 30-day mortality and delirium, respectively [40]. In a prospective study [38], NSE and S-100 β-protein levels were measured to evaluate cerebral injury, and to predict the outcome in severe sepsis; they were found to be higher in the most severe patients. When compared to NSE, S-100 β-protein was a stronger predictor of survival, and S-100 β-protein levels above 4 μg/L were associated with severe brain ischemia or hemorrhage.

Recently, the amino-terminal propeptide of the C-type natriuretic peptide (NT-proCNP) has been compared to NSE and S-100 β-protein in both septic patients and non-septic controls [41], as a biomarker for SAE and neuroinflammation. The plasma levels of NT-proCNP and S-100 β-protein were higher in the septic than in the non-septic group on day 1, and decreased over time, while no differences were observed for the NSE levels in both groups. The cerebrospinal fluid (CSF) levels of NT-proCNP, NSE and S-100 β-protein were also compared with the CSF IL-6 levels. A correlation between the CSF NT-proCNP and CSF IL-6 levels was found, suggesting a link between neuro- and systemic inflammation in sepsis. However, larger prospective trials are required to confirm these preliminary observations.

Finally, in a prospective observational study, light (NfL) and heavy (NfH) neurofilament chain levels, both in CSF and plasma, were measured as biomarkers of neuroaxonal injury in septic and non-septic patients [42]. This study showed that the plasma NfL levels were higher in septic than in non-septic patients; they were significantly higher in patients with SAE, and correlated with the severity of SAE and a poor functional outcome. The same findings were reported for the NfL levels in the CSF, further showing a correlation between the higher value of CSF NfL levels and mortality.

### 4.3. Neuroimaging

Patients with a sudden alteration in mental status may often require neuroimaging to exclude some potential life-threating conditions, such as ischemic stroke or diffuse cerebral hemorrhage. Neuroimaging of septic patients presenting with an alteration in their mental status is often unremarkable. Although computed tomography (CT) scan is the first-step tool to detect neurological deficit, this technique is not as sensitive as magnetic resonance imaging (MRI) to show neuropathological findings [37].

In patients with SAE, MRI demonstrates heterogenous and non-specific patterns of brain injury, such as ischemic lesions, diffuse leukoencephalopathy, severe vasogenic edema, atrophy, white matter hyperintensities, and cortical and subcortical hemorrhage [30,37]. In survivors of sepsis, atrophy and white matter changes on MRI have been associated with ICU-acquired delirium and worse cognitive outcomes at 1-year follow-up [30,43].

However, in the case of acute changes in mental status, especially in the presence of focal signs, a CT scan remains the imaging of choice to rule out ischemic or hemorrhagic brain injury. In a case of persistent encephalopathy, when the source of the infection has been successfully controlled and the major confounding factors have been excluded, brain MRI should be considered [12].

Treating SAE is as equally challenging as its diagnosis. Because there is no specific treatment for SAE, clinical management relies on early detection and control of the infection source, together with general measures oriented around hemodynamic stability, in order to optimize organ and cerebral perfusion [10,12]. Supportive care of patients with SAE includes symptomatic management of delirium and EEG monitoring [4]. The latter is useful to detect seizures that would be unrecognized in these patients, as seizures are often non-convulsive in this population [32,33].

The identification of potential modifiable factors, such as acute renal failure and common metabolic disturbances (hypo-hyperglycemia and hypernatremia), may decrease the risk of developing SAE [44].

## 5. Sepsis-Associated Long-Term Cognitive Impairment

In the last two decades, the overall survival rate of patients admitted to the ICU has increased; consequently, medical research is also oriented around the long-term effect of critical illness. In particular, numerous studies have tried to evaluate the impact of critical illness on functional and cognitive outcomes. In a large multicenter prospective cohort study, Pandharipande et al. [45] studied a general population of critically ill patients to quantify the incidence of cognitive impairment 3 and 12 months after ICU discharge, and to investigate any potential risk factors associated with the stay in ICU. The study revealed that the patients’ global cognition scores were 1.5 below the general population mean, and that they were similar to those of patients with mild cognitive impairment. In particular, at 3 months, 40% of the patients had worse global cognition scores than those observed after a moderate traumatic brain injury, while 26% had similar scores to those of patients with mild Alzheimer’s disease. This impairment was also confirmed after 12 months, meaning that cognitive impairment tends to persist over a long period. Moreover, the same study found that cognitive impairment after critical illness was not limited to older people, and that the global cognition scores were low regardless of the burden of pre-existent comorbidities. The study conducted by Pandharipande et al. included a general ICU population of patients admitted to the ICU, regardless of their initial diagnosis; however, 30% of the patients had an initial diagnosis of sepsis, showing that even after the acute event, patients who suffered from severe sepsis demonstrated functional and cognitive impairment in the long term.

In order to study sepsis and septic shock as independent variables associated with long-term cognitive impairment, other studies were conducted. In particular, a prospective cohort study [46], by Iwashyna, enrolled survivors of severe sepsis, in order to determine the change in cognitive impairment and physical functioning, compared to their pre-sepsis state, with each patient serving as his/her own control. The authors found that an episode of severe sepsis was associated with clinically significant progression to moderate or severe cognitive impairment, and that this result was not associated with the presence of pre-existing cognitive impairment. In addition to this, the long-term negative effects of sepsis were related not only to cognition, but also to functional outcomes, since the patients with better baseline physical functioning showed worse outcomes. Another important finding by Iwashyna was that the changes in functional and cognitive outcomes after severe sepsis were worse than those observed after general hospital admission, meaning that severe sepsis may be an independent risk factor for the development of long-term cognitive and functional impairment.

The temporal window for long-term cognitive impairment has been investigated, but with contradictory results. Calsavara et al. [47] evaluated septic patients at ICU discharge and after 12 months, reporting early impairment in MMSE and constructional praxis tests immediately after ICU discharge, with partial improvement at 12 months. In contrast with these findings, a prospective study conducted by Wang et al. [48] showed no acute decrease in cognition after sepsis, but faster cognitive decline in the long term. In particular, this study found that, compared to a control population, cognitive decline was accelerated by seven times, and that the odds of cognitive impairment were increased by 8% each year after a septic event. Despite the fact that the two studies have divergent results about the immediate post-acute phase of sepsis, both of them confirm the presence of long-term cognitive decline that persists, even after 12 months.

Interestingly, sepsis-associated long-term cognitive and functional impairment is also subjectively perceived by patients. Konig et al. [49] interviewed 15 patients of different ages who underwent an ICU admission for severe sepsis. In the interviews, the patients reported a sudden and relevant deterioration in health and functioning after ICU discharge, refusing to accept their state after critical illness as “normal”, compared to their previous state. Concerning their cognitive state, many of them demonstrated symptoms such as a lack of concentration, loss of memory, speech disturbance and disorientation. Moreover, they appeared to be anxious and to suffer from psychological impairment, with nightmares, flashbacks and memories of “coma dreams”, related to their period in the ICU.

All these studies confirm the association between an episode of severe sepsis and the development of long-term cognitive and functional impairment. However, one limit of all the research on this topic is the absence of a uniform and validated set of cognitive and behavioral tests for the evaluation of patients. Moreover, the timing of long-term evaluation appears to be different in all the studies, and, in some of them, the evaluation was not carried out by trained personnel, such as neurologists or psychiatrists. All these problems tend to make the analysis of the overall results more difficult, and show the need for a consensus about the tools and timing of the evaluation of sepsis-related long-term consequences. Moreover, the definition of sepsis itself affects the rate of long-term cognitive impairment, since the Sepsis-3 definition for sepsis hospitalization is linked to a trajectory of cognitive decline that is twice as high as that observed with the Sepsis-SIRS definition [48].

The findings about sepsis-related long-term cognitive impairment have led to all the possible correlations with clinical and laboratory data being searched, which could be useful to define an early diagnosis. Correlations between the neuroimaging findings and long-term cognitive impairment in patients undergoing ICU admission were found. In particular, it was demonstrated that the presence of smaller total brain volumes and greater brain atrophy, shown on MRI at 3 months after ICU discharge, were associated with worse cognitive performances at 12 months, and smaller thalamic and cerebellar volumes were linked to worse executive functioning [43]. These results confirmed the presence of an organic substrate in long-term cognitive impairment caused by critical illness. Semmler et al. [5], in particular, investigated a population of septic patients and their changes in brain imaging after severe sepsis, comparing this subpopulation to the general population of ICU survivors. They found that both the brain MRI and EEG of septic patients were worse than those of the general ICU population, with a larger reduction in hippocampus volume and more frequent EEG dysfunction in the first group.

Additionally, Calsavara [47] found a correlation between long-term cognitive impairment and some laboratory data collected during the acute septic event. In particular, a negative correlation was found between the serum concentration of inflammatory mediators (TNF, IL-6, IL-4 and IL-10) and worse scores in neurocognitive tests.

The correlation between sepsis-associated delirium and long-term cognitive outcome has also been investigated. Delirium is defined as an acute form of cerebral dysfunction, with a change or fluctuation in baseline mental status, inattention, and either disorganized thinking or an altered level of consciousness [50]. The occurrence of delirium is frequent in ICU patients, and its duration is an independent predictor of cognitive impairment, both in the early phase and in the long term [51]. This association is independent from sedative or analgesic medication use, age, pre-existing comorbidities, ongoing conditions and the duration of mechanical ventilation [45,51]. In most of the previous studies performed in the ICU, a large number of septic patients were included to investigate the link between delirium and long-term cognitive impairment [52].

Previous treatment guidelines regarding sepsis focused on short-term outcomes and mortality. The Surviving Sepsis Campaign 2021 guidelines [1] contributed to the definition of the long-term neuropsychological outcomes in sepsis survivors, although no recommendation was given about this topic because of insufficient evidence. So far, no large studies have been performed to investigate which kind of measures are needed to avoid cognitive impairment; moreover, there is only little evidence, aimed at improving long-term outcomes, about possible changes in the acute treatment of severe sepsis and septic shock [53].

Although data about septic patients admitted to the ICU are limited, it is important to underline that this subgroup of patients needs to be treated with all the measures already known to improve long-term outcomes after ICU admission (Figure 2). Several studies found different ways to improve the ICU daily care routine. Vincent et al. [54] proposed a model summarized by the acronym eCASH, which includes a bundle of interventions aimed at managing patient comfort with minimal sedation, optimal analgesia and maximal human care. This approach emphasizes the optimization of pain management to avoid deep sedation, which is one of the most important risk factors for the development of delirium, with attention given to early mobilization and sleep promotion. In addition, this model aims to improve the engagement of a patient’s family, with measures such as the open ICU concept and the implementation of the active presence of family and visitors in the daily ICU routine. These measures are at the base of a new patient-centered concept of ICU, which aims to prevent neurological, emotional and functional impairment by improving the daily care routine.

## 6. COVID-19

Regarding long-term cognitive impairment after severe sepsis, a new important concern has arisen with the SARS-CoV-2 pandemic. This respiratory viral infection may impair outcomes in survivors, not only because ICU admission is needed in up to one-quarter of hospitalized patients, but also for the viral neurotropism. It has been demonstrated that SARS-CoV-2 infection can be associated not only with respiratory and flu-like symptoms, but also with neurological manifestations, defined as Neuro COVID. This condition includes a wide number of clinical features, both central and peripheral, such as an altered mental status, a neuropsychiatric diagnosis, and the development of Guillain–Barrè syndrome or peripheral myelitis, [55] based on different mechanisms, such as cerebrovascular accidents (both hemorrhagic and ischemic), encephalomyelitis or demyelinating syndromes. All these manifestations may be due to the direct effect of the virus on the nervous system, and to the host’s immunological response, together with consequential inflammatory hyperactivation [55,56].

After the acute phase, patients affected by COVID-19 infection may also experience a systemic disease that lasts for four or more weeks following the acute phase of infection. This condition, known as Long COVID, may present severe fatigue, breathlessness, neurocognitive deficits, headaches, joint pain, and a decline in the quality of life [57,58,59]. This clinical picture may be due to different factors, related to both hospitalization, such as post-intensive care syndrome (PICS) or social isolation, associated with the specific COVID-19 infection, and to the infection itself, such as residual inflammation or viral organ damage [58].

In this setting, neurocognitive impairment represents one of the most important factors affecting recovery, with patients experiencing difficulties in attention, concentration, memory, chronic fatigue, insomnia, depression and anxiety [58]. Carfì et al. described that two months after discharge from the ICU, about 44% of COVID-19 patients showed a worsened quality of life, with more than 50% of them showing fatigue [60]. In a quality improvement survey, Razai found that post-acute COVID-19 patients self-reported neurocognitive difficulties, such as a lack of memory and concentration, headache and “brain fog” [61].

Interestingly, long COVID sequelae seem to not only affect hospitalized patients, as non-hospitalized patients also claim to experience impairments in functional recovery, cognition and quality of life [62]. This suggests that COVID-19 cognitive consequences may not only be limited to the most severe forms of infection. In hospitalized patients, especially in those who face ICU admission, the onset of long-term neurocognitive symptoms may be worse because of factors such as the hospitalization itself, isolation and ICU-related risk factors, such as analgesia and sedation, prolonged mechanical ventilation, and the onset of delirium.

In this perspective, once again, the key to preventing long-term neurocognitive impairment seems to be improving all the clinical modifiable factors related to ICU admission, such as avoiding deep sedation regimens, granting an effective analgesic plan, enhancing early mobilization, and trying to allow family participation and open ICU models. In addition to the acute-phase treatment, another fundamental factor to consider is the importance of a comprehensive follow-up in COVID-19 patients, even after hospitalization, in order to promptly recognize and treat any neurological, psychological or functional alterations. This kind of follow-up can be performed using all the validated scales that are commonly used for the general population, such as the Short Form Health Survey 36 (SF-36) for evaluation of the overall health status, and the Mini-Mental Test for evaluation of the neurocognitive and functional state [58].

## 7. Conclusions

Short- and long-term neurological sequelae after severe infection have been widely investigated. The pathophysiology of SAE is multifactorial and its clinical diagnosis is not supported by specific diagnostic instruments, thus making its recognition difficult. SAE begins as a reversible complication during sepsis, but it may evolve into an irreversible long-term cognitive impairment, affecting functional abilities, verbal learning and memory, thus leading to a reduction in the quality of life and the ability to return to work in ICU survivors. Therefore, in the absence of specific therapies, general treatment principles of sepsis, avoidance of neurotoxic agents and metabolic derangement should be implemented, together with early physical rehabilitation, in order to reduce the risk of subsequent neurodegeneration.

## Figures and Tables

**Figure 1 biomedicines-10-00526-f001:**
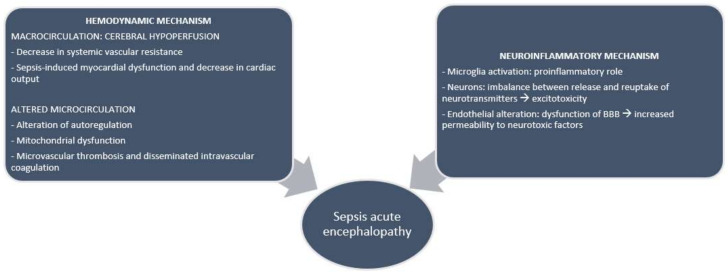
Pathophysiology of sepsis-associated encephalopathy.

**Figure 2 biomedicines-10-00526-f002:**
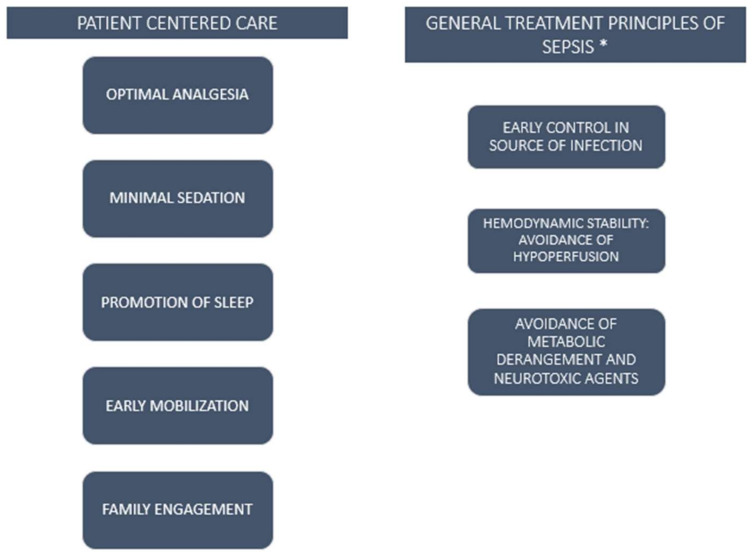
Strategies to prevent sepsis-induced long-term cognitive impairment. * according to recent guidelines.

## Data Availability

Not applicable.

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
