# Peer review of "Neuropsychological Outcome of Critically Ill Patients with Severe Infection"

_biomedicines, 2022, doi:10.3390/biomedicines10030526_

Round 1
Reviewer 1 Report
It is a good review on neuropsychological outcome of critically ill patients with severe infection. It Sepsis associated encephalopathy is a severe brain dysfunction which may contribute to long-term cognitive impairment. Two mechanisms are involved in its pathophysiology: neuroinflammation and hemodynamic disturbances. Diagnosis is difficult because of the absence of specific radiological and biological criteria. There are not specific therapies, general treatment principles of sepsis, avoidance of neurotoxic agents and metabolic derangement should be implemented together with an early physical rehabilitation in order to reduce the risk of subsequent neurodegeneration. Overall, the article provides a lot of information about this topic, is interesting and valuable to readers, and may be published in the Biomedicines journal with minor revisions:
Check the text there are some erratum such as: sydnromes (page 7; lines 33-34)
Author Response
Thank you for your comment and your notes to our article!
I checked again the text and found the erratum you mentioned and two other misspelled words.
Reviewer 2 Report
This article highlights the severity of the neurological complications that can be acquired by critically ill patients with septic shock (encephalopathy associated with sepsis) and the importance of combating them from the beginning of the onset of symptoms.
Sepsis-associated encephalopathy is a complication of septic shock that is reversible in the first phase, then long-term becomes irreversible, significantly reducing the quality of life.
As a novelty, the article brings a change in the approach of the patient with septic shock, namely increasing the attention to neurological complications and combating encephalopathy associated with sepsis from the beginning not through a specific treatment, but by approaching the critical intensive care patient using centered care. per patient (optimal analgesia, minimal sedation, improved sleep, early mobilization and higher family involvement for a better psyche) and general treatment of sepsis (control and etiological treatment of the source of infection, maintaining hemodynamic stability and avoiding imbalances metabolic and neurotoxic agents).
Also, even if sepsis-associated encephalopathy does not benefit from routine paraclinical investigations that could provide an early diagnosis, the determination of neuron-specific enolase (NSE), S-100 beta-protein, NT-proCNP natriuretic peptide and CSF IL -6 can diagnose encephalopathy. This study found that the biomarkers mentioned have a higher value in sepsis-associated encephalopathy than in other types of encephalopathies, but there are studies with contradictory results regarding the correlation between the increased value of these biomarkers and the severity of encephalopathy. Of these, the S-100 protein was the strongest predictor of survival, with low values being associated with a low mortality rate. I
In addition to biomarkers, the use of MRI has benefits in the early diagnosis of encephalopathy?
Author Response
Thank you for your comment and your notes to our article!
Concerning the use of MRI in the early diagnosis of encephalopathy, it must be said that because of the lack of a specific sign related to SAE and the presence of a wide range of radiological presentations, MRI is not able to detect a specific pattern that could be sensitive and specific for sepsis associated encephalpathy, both in the acute phase and in the long term. As our article wants to underline, SAE is a condition difficult to diagnose because of the lack of specific radiological and biological criteria.